

# Uncertainty estimation for a new exponential filter-based long-term root-zone soil moisture dataset from C3S surface observations

Adam Pasik[1], Alexander Gruber[1], Wolfgang Preimesberger[1], Domenico De Santis[2], and Wouter Dorigo[1]

[1]Department of Geodesy and Geoinformation, TU Wien, Wiedner Hauptstraße 8, 1040 Vienna, Austria
[2]Research Institute for Geo-Hydrological Protection, National Research Council, Via della Madonna Alta 126, 06128 Perugia, Italy

**Correspondence:** Adam Pasik (adam.pasik@geo.tuwien.ac.at)

**Abstract.** Soil moisture is a key variable in monitoring climate and an important component of the hydrological, carbon, and energy cycles. Satellite products ameliorate the sparsity of field measurements but are inherently limited to observing the near-surface layer, while water available in the unobserved root zone controls critical processes like plant water uptake and evapotranspiration. A variety of approaches exists for modelling root-zone soil moisture (RZSM), including approximating it from surface layer observations. While the number of available RZSM datasets is growing, they usually do not contain estimates of their uncertainty. In this paper we derive a long-term RZSM dataset (2002–2020) from the Copernicus Climate Change Service (C3S) surface soil moisture (SSM) COMBINED product via the exponential filter (EF) method. We identify the optimal value of the method's model parameter $T$, which controls the level of smoothing and delaying applied to the surface observations, by maximizing the correlation of RZSM estimates with field measurements from the International Soil Moisture Network (ISMN). Optimized T-parameter values were calculated for four soil depth layers (0–10 cm, 10–40 cm, 40–100 cm, and 100–200 cm) and used to calculate a global RZSM dataset. The quality of this dataset is then globally evaluated against RZSM estimates of the ERA5-Land reanalysis. Results of the product comparison show satisfactory skill in all four layers with median Pearson correlation ranging from 0.54 in the topmost to 0.28 in the deepest soil layer. Temporally-dynamic product uncertainties for each of the RZSM product layers are estimated by applying standard uncertainty propagation to SSM input data and by estimating structural uncertainties of the EF method from ISMN ground reference measurements taken at the surface and in varying depths. Uncertainty estimates were found to exhibit both realistic absolute magnitudes as well as temporal variations. The product described here is, to our best knowledge, the first global, long-term, uncertainty-characterized, and purely observation-based product for RZSM estimates up to 2 m depth.

## 1 Introduction

Soil moisture (SM) is an essential climate variable (ECV) crucial for understanding and modelling the Earth's climate, and an important control of hydrological, energy, and carbon fluxes (GCOS, 2022; Dorigo et al., 2021a). Global monitoring of SM is necessary for a variety of applications such as meteorological modelling (Albergel et al., 2008), monitoring drought (Tobin et al., 2017), and modelling groundwater recharge (Bouaziz et al., 2020), runoff, and catchment response to storms (Brocca et al., 2010).



In situ SM measurements are considered to provide the most accurate SM data but can differ greatly in measuring equipment and usually lack estimates of their uncertainties (Dorigo et al., 2011). Widely distributed SM field measurements are available from centralized platforms such as the International Soil Moisture Network (ISMN) (Dorigo et al., 2021b). While being essential for satellite and model product calibration and validation, in situ measurements lack the spatial coverage necessary for large-scale applications, especially in the Global South (see Figure A1; Dorigo et al. (2021a); Mishra et al. (2020)). Quasi-

global SM information is available from modelled and satellite products, but their spatial resolution is very coarse (usually tens to hundreds of square kilometers) and usually insufficient to resolve the significant spatio-temporal heterogeneity of SM, which poses challenges to large-scale monitoring (Brocca et al., 2010). Global land surface model products provide gap-free and long-term SM estimates at various depths and chosen time intervals, but are computationally expensive and may depend on many auxiliary inputs that are not always available globally or in sufficient quality or resolution (Mishra et al., 2020; Albergel

et al., 2008). In contrast, remote sensing retrievals are available only at satellite overpass times and unreliable under various conditions including frozen ground, dense vegetation, or radio frequency interference (RFI) (Gruber et al., 2019; Dorigo et al., 2017). Moreover, microwaves used for SM retrieval contain mainly information on water content in the surface layer, hampering their usability for studying or modelling processes in the soil root zone. Root-zone soil moisture (RZSM), often defined as the water present in the top meter of the soil column (Mishra et al., 2020; Baldwin et al., 2017; de Lange et al., 2008), is a com-

ponent of the Global Climate Observing System (GCOS) ECV portfolio and a necessary variable for closing the water cycle (GCOS, 2016, 2022). RZSM also represents the water available for plant water uptake and thus affects evapotranspiration rates (Martens et al., 2017; Ford et al., 2014; Albergel et al., 2008) and plays a critical role in agricultural productivity forecasting (Wang et al., 2017) and drought monitoring (Vreugdenhil et al., 2022; Tobin et al., 2017).

    The existing link between SM dynamics in the surface layer and the root zone (Albergel et al., 2008; Wang et al., 2017;

Ford et al., 2014; Sure and Dikshit, 2019) allows for estimating RZSM from surface SM (SSM) observations via a variety of hydrological models (Manfreda et al., 2014; Bouaziz et al., 2020), including reanalysis (Muñoz Sabater et al., 2021; Rodell et al., 2004) and data assimilation techniques (Reichle et al., 2017). An alternative, less complex approach that approximates RZSM solely from SSM estimates—and can thus be readily applied to satellite retrievals—is the so-called exponential filter (EF) method (Wagner et al., 1999; Albergel et al., 2008). In essence, the EF method approximates conditions in the root zone

by smoothing and delaying SSM, which is generally characterized by greater fluctuations (Beck et al., 2009; Mahmood and Hubbard, 2007). Even though the coupling strength between the surface and root-zone layers decreases with depth (Mahmood and Hubbard, 2007; Ford et al., 2014; Mishra et al., 2020) and the skill of the method in predicting RZSM has been demonstrated to deteriorate accordingly (Paulik et al., 2014; Brocca et al., 2010; Sure and Dikshit, 2019), it is still widely used due to its relatively good performance and independence of ancillary inputs as well as its low computational cost and overall simplic-

ity. However, the EF method is susceptible to prolonged data gaps in SSM data and thus requires an adequate number of input observations within a time interval consistent with the temporal scale of RZSM dynamics.

    Regardless of the method used to derive RZSM estimates, most products do not provide information about the magnitude of random errors such as the standard deviation of their distribution, hereinafter referred to as uncertainties (Gruber et al., 2020). Two approaches have been proposed to characterize the time-variant quality of RZSM estimates derived with the EF method.



The first approach, reported in Bauer Marschallinger (2018) and also utilized in this study, is a quality flag that is derived from the number of valid SSM estimates available within a specific time window preceding a specific EF-based RZSM estimate. The second approach, proposed by De Santis and Biondi (2018), uses the standard law of uncertainty propagation (Taylor, 1997) in order to characterize the random error variances of EF-based RZSM estimates. This approach takes into account the uncertainties of both the SSM input data and the EF model parameter, but does not consider the model structural error (Beven,

2005). The latter, due to the simplistic nature of the EF method and the limited surface–root zone coupling, can also contribute significantly to the uncertainty budget and thus must not be neglected when characterizing product errors.

In this paper, we propose to estimate the model structural uncertainty of the EF using in situ measurements of surface and root-zone SM from the ISMN. We then use these estimates together with the law for the propagation of uncertainties (similar to De Santis and Biondi (2018)) to produce a global, fully error-characterized RZSM data set for four soil layers (0–10 cm,

10–40 cm, 40–100 cm, and 100–200 cm) between 2002 and 2020, taking C3S soil moisture as input to the model. To our best knowledge, this is, as yet, the longest available observation-based, error-characterized global RZSM product.

## 2 Datasets and data pre-processing

### 2.1 C3S surface soil moisture

Global input satellite surface observations were obtained from the Copernicus Climate Change Service (C3S) Surface Soil

Moisture COMBINED product v202012, hereinafter referred to as C3S SSM. C3S SSM is a merged product that combines satellite SSM retrievals from four active and ten passive microwave sensors into a daily global dataset on a regular 0.25 degree grid, expressed in volumetric units ($m^3/m^3$) (C3S, 2020). Invalid retrievals due to frozen ground, dense vegetation, RFI, and other factors are masked out. Although the C3S product provides SSM data from 1978 onward, their quality and spatio-temporal coverage increases significantly in more recent periods when sensors measuring in frequency domains better suitable

for SSM retrieval are available. Therefore, only C3S SSM data for the period 2001-2020 were used in this study. Note that data from the first year of this period was used only as the model adjustment period and not included in later analyses.

The uncertainty estimates provided for the merged SSM retrievals in the C3S SSM product were computed by means of Triple Collocation Analysis (TCA) (Gruber et al., 2017). More specifically, (stationary) uncertainties were estimated for each satellite sensor separately and used to calculate the merging weights. Uncertainties of the merged SSM estimates were then

calculated from the law for the propagation of uncertainties to account for the quality improvement due to the merging. Note that the distinctive life spans of the used satellite missions, therefore, lead also to distinctive changes in the data quality of the merged product. These sudden changes in product uncertainty are hereinafter referred to as structural breaks (Preimesberger et al., 2021). As more and newer sensors provide better-quality retrievals, mean uncertainty values after each structural break typically decrease (Gruber et al., 2017). This is apparent, e.g., in the shift in C3S SSM uncertainty values after the introduction

of AMSR-E in 2002 (van der Schalie et al., 2017; Gruber et al., 2019).



C3S data are readily available from the Copernicus Climate Data Store (CDS) and detailed information on the C3S dataset and its underlying ESA CCI v5 merging algorithm can be found in the relevant documentation (C3S, 2020; Dorigo et al., 2021c).

## 2.2 Soil moisture field measurements

Field measurements for optimizing the model parameters of the EF method and for estimating its uncertainties were obtained from the International Soil Moisture Network (ISMN) for the period 2002-2020 (Dorigo et al., 2021b). Only data from sensors with a measuring depth $\leq$ 200 cm and internally flagged as reliable (Dorigo et al., 2013) were considered. Measuring depths of SM sensors placed vertically in a depth range, e.g., 10–40 cm, refer to their mean measuring depth. Data from multiple sensors installed at the same location and depth were averaged. ISMN data, typically available as hourly readings, were aggregated to

mean daily values to match the temporal sampling of satellite observations. Furthermore, we only used ISMN stations where at least 100 data points concurrent with C3S SSM retrievals were available. Notably, approximately 80% of selected ISMN time series originate from North America and Europe (Figure A1) and the availability of data declines with depth.

## 2.3 ERA5-Land soil moisture

ERA5-Land (E5L) is a multi-decadal climate reanalysis with an extensive portfolio of land variables computed by the assimi-

lation of ERA5 atmospheric variables into the H-TESSEL land surface model (Muñoz Sabater et al., 2021). Modelled SM data are available for four depth layers (0–7 cm, 7–28 cm, 28–100 cm, and 100–289 cm) on a regular 0.1 degree grid and accessible via the Copernicus Climate Data Store (CDS) (Muñoz Sabater, 2019, 2021). We used E5L for a product intercomparison with the RZSM product developed in this study, carried out for the period 2002–2020 within the Quality Assurance for Soil Moisture framework (QA4SM; https://qa4sm.eu), which automatically resamples and matches observations of the compared

datasets and delivers a wide range of validation metrics.

## 3 Methods

### 3.1 Exponential filter

The EF method (Wagner et al., 1999) relies on a simple two-layer water balance model where the only considered exchange between the surface layer and the reservoir below it is infiltration. The method assumes that the fluxes from the surface to the

sub-surface layers are proportionate to the difference in SM content between both layers. In this study, we utilize the recursive formulation of the method (Albergel et al., 2008):

$$RZSM(t_n) = RZSM(t_{n-1}) + K_n \cdot (SSM(t_n) - RZSM(t_{n-1})) \tag{1}$$

$t_n$ and $t_{n-1}$ denote timestamps (in days) of the current and previous SSM observations, respectively. Conditions in the root zone are approximated by a weighted combination of the new input SSM observation and past model estimates, with more





recent estimates receiving higher weights on a time scale defined by the method's only parameter $T$ (temporal length, typically in days). Weights are controlled by the gain term $K$, which ranges from 0 to 1 and is calculated as follows:

$$K_n = \frac{K_{n-1}}{K_{n-1} + e^{-\frac{t_n - t_{n-1}}{T}}} \qquad (2)$$

At initialization, when no preceding estimates are available, the EF calculation is started with $K_0 = 1$ and $RZSM(t_0) = SSM(t_0)$.

Temporal variability in the root zone is generally smaller than at the surface, hence the $T$-value and its associated level of smoothing applied to the SSM data increase with depth (Wagner et al., 1999; Paulik et al., 2014; Wang et al., 2017; Beck et al., 2009; Mahmood and Hubbard, 2007). The optimal $T$-value ($T_{opt}$, the value that leads to the best possible representation of RZSM at a certain location using the EF), has been related to differences in utilized SSM sensors (Bouaziz et al., 2020; Sure and Dikshit, 2019), SSM sampling frequency (Brocca et al., 2010; Pellarin et al., 2006), and land surface features (Albergel
et al., 2008; de Lange et al., 2008). In particular, $T$ acts as a conglomerate proxy for various environmental factors assumed to rule the infiltration process (e.g., soil texture, evapotranspiration, and climate), but past research on the importance of the exact driving factors is inconclusive and even contradictory (Wang et al., 2017; Bouaziz et al., 2020). To optimize the $T$-parameter, numerous control factors have been tested (Bouaziz et al., 2020; Mishra et al., 2020; Stefan et al., 2021) and ever more sophisticated methods been employed, including machine learning approaches (Grillakis et al., 2021).

Due to the high spatio-temporal heterogeneity of SM (Famiglietti et al., 2008) and its surface–root zone coupling—and hence the difficulty in properly estimating the $T$-parameter accurately—an uncalibrated value of $T{=}20$ has sometimes been used to describe all of the water content in the first 100 cm of the soil column (Wagner et al., 1999; de Lange et al., 2008). Results obtained by using a constant value $T{=}20$ were similar to those obtained with $T$-values calibrated for soil texture (de Lange et al., 2008). Limited sensitivity of the EF to $T$ due to different environmental factors was also observed by other studies,
which supports choosing a single value for $T_{opt}$ to represent a particular depth for large areas or even globally (Albergel et al., 2008; Brocca et al., 2010, 2011; Grillakis et al., 2021).

### 3.1.1 RZSM quality flags

Prolonged temporal data gaps will cause $K$ to increase and may cause the EF to put excessive weight on new SSM input. In the extreme case, a very long data gap (whose duration depends on the chosen $T$-value) can reset the EF to the initial state of
$K_n = 1$ and $RZSM(t_n) = SSM(t_n)$ (see above). We run a one-year adjustment period (2001) for $K$ to reach an equilibrium state, and utilize the EF quality flag ($qflag$) described in Bauer Marschallinger (2018) to avoid such re-initializations due to frequent and/or persistent data gaps. The $qflag$ is recursively calculated for each RZSM estimate and reflects the availability of SSM input data in the preceding time period.





$$qflag(t_n) = \begin{cases} 1 + qflag(t_{n-1}) \cdot e^{-\frac{t_n - t_{n-1}}{T}}, & \text{if SSM at } t_n \text{ is available} \\ qflag(t_{n-1}) \cdot e^{-\frac{t_n - t_{n-1}}{T}}, & \text{if SSM at } t_n \text{ is unavailable} \end{cases} \tag{3}$$

The quality flag calculation is initialized with $qflag(t_n) = 1$. A normalization factor of $\sum_{j=0}^{\infty} e^{-\frac{j}{T}}$ is used to express the calculated flag values in percentages with higher values indicating a greater density of SSM data available for calculation. If the quality flag falls below a $T$-specific threshold, RZSM estimates are masked out. The thresholds used here have been interpolated from those empirically determined by Bauer Marschallinger (2022) for a set of discrete $T$-values (35%, 40%, 45%, 50%, 55%, 60%, 65%, and 70% for the $T$-values 2, 5, 10, 15, 20, 40, 60, and 100, respectively). If input data is unavailable but
satisfactory data density has been achieved in preceding days, the latest RZSM estimate is propagated forward until new input data becomes available or the quality flag drops below its respective threshold. In the latter case, the output value is masked out. Importantly, even if new SSM input becomes available to the EF after prolonged data gaps, RZSM estimates derived from it remain masked until the $qflag$ exceeds the aforementioned threshold again.

### 3.1.2   *T*-parameter optimization

We optimize $T$ for a particular depth of the soil column by maximizing the correlation between the satellite-based RZSM estimates and the in situ measurements (Paulik et al., 2014; Grillakis et al., 2021). Satellite and in situ data are matched in space by means of the nearest neighbour method. The impact of the spatial mismatch error between the large footprint of the satellite-based product and point-scale field measurement is mitigated by excluding time series that exhibit a correlation coefficient (Pearson's r) lower than 0.5 (Grillakis et al., 2021) or are not statistically significant ($p \geq 0.05$).

EF calculations are repeated for $T$-values 1–100 and $T_{opt}$ is selected for each of the available ISMN time series based on the highest correlation coefficient. We then group $T_{opt}$ values based on the measurement depth of the respective in situ sensor into four bins corresponding to the RZSM target layers. These depth layers, chosen to be 0–10 cm, 10–40 cm, 40–100 cm, and 100–200 cm, were defined to reflect those in common model-based RZSM products (Rodell et al., 2004; Muñoz Sabater et al., 2021). Finally, the median value of $T_{opt}$ from each bin is chosen to compute a global RZSM product from the C3S SSM
dataset.

A cross-validation is carried out to verify that $T_{opt}$ values were not over-fitted to the local ISMN site conditions. Therefore, the sample set is randomly divided into 5 subsets of equal size (per bin), then each of the subsets was used once to validate the method fit to the remaining 4 bins.

### 3.2   Uncertainty estimation

### 3.2.1   Baseline method

In De Santis and Biondi (2018), the standard law for the propagation of uncertainties is applied to the EF method. We use this approach as a baseline for our analyses. The recursive formulation of this baseline method is as follows:



$$\sigma(RZSM_n) = \sqrt{\Delta_n^2 + \left(\frac{\partial RZSM_n}{\partial T}\right)^2 \sigma^2(T)} \tag{4}$$

where:

$$\Delta_n^2 = K_n^2 \sigma^2(SSM_n) + (1 - K_n)^2 \Delta_{n-1}^2 \tag{5}$$

and:

$$\frac{\partial RZSM_n}{\partial T} = \frac{K_n}{T}\left[G_n(RZSM_{n-1} - RZSM_n) + e^{-\frac{t_n - t_{n-1}}{T}}\frac{T}{K_{n-1}}\frac{\partial RZSM_{n-1}}{\partial T}\right] \tag{6}$$

with $G_n$ defined as:

$$G_n = e^{-\frac{t_n - t_{n-1}}{T}}\left(G_{n-1} + \frac{1}{K_{n-1}}\frac{t_n - t_{n-1}}{T}\right) \tag{7}$$

$\sigma(RZSM)$ and $\sigma(T)$ denote the uncertainty of the RZSM estimates and the EF model parameter $T$, respectively. The equation is initialized as $\Delta_0 = \sigma(SSM_0)$, $\partial RZSM_0 / \partial T = 0$ and $G_0 = 0$. Uncertainties of the SSM input data are considered by the $\Delta$ term, which also takes into account the effect of possible prolonged input data gaps dependent on the $T$-value. The Jacobian term $\partial RZSM/\partial T$ assumes high values proportional to the latest SSM input variability on a time scale related to the $T$-parameter. This is reflected in significant changes in the RZSM value associated with wetting or drying of the soil.

### 3.2.2  $T$-parameter uncertainty

De Santis and Biondi (2018) used an arbitrary value of $\sigma(T)$ equal to 10% of locally calibrated $T_{opt}$. This is in line with other studies on SM uncertainty propagation (Parinussa et al., 2011; Pathe et al., 2009), who used this uncertainty percentage for parameters without well-defined accuracy. In our study, we determine $T_{opt}$ values based on a limited number of available in situ time series and apply these values to estimate RZSM globally. Consequently, $\sigma(T)$ is likely to be greater due to a variety of environmental conditions not accounted for or underrepresented in the available in situ sample. We therefore propose the median absolute deviation (MAD) of $T_{opt}$ (Section 3.1.2) as a more appropriate proxy for $\sigma(T)$. In this case, the MAD is preferred over the variance because the sampling distribution of $T_{opt}$ is both non-Gaussian and bounded (Leys et al., 2013).

### 3.2.3  EF model structural uncertainty

Recall that the standard law for the propagation of uncertainty (which is used in the baseline method) does not account for model structural uncertainty in the EF, which, due to the simplistic nature of the method and the limited surface–root zone coupling, can account for a significant portion of the overall uncertainty budget.

We propose to estimate model structural uncertainty ($\sigma(EF)$) from in situ data using stations that operate sensors both at the surface and in the root zone. At these stations, we derive RZSM estimates from the SSM measurements using the EF method and then compare them to actual RZSM station measurements. For this analysis, the $T$-value was optimized for each station and depth individually to minimize its influence on the estimation of $\sigma(EF)$. This provides direct estimates for $\sigma(EF)$ as:



$$\sigma(EF) = ubRMSD(RZSM_{EF}, RZSM_{ISMN}) \tag{8}$$

where $ubRMSD$ denotes the unbiased root-mean-square difference. Note that 'unbiased', in this case, does not only refer to a correction for bias in the mean (as is most commonly done) but also to a correction for bias in variance, which also constitutes an unintended systematic component in the $RMSE$ (Gupta et al., 2009). Only sites with measurements from more than a single
depth and at least one sensor within the surface layer ($\leq 10$ cm) were selected. Time series with negative correlation between EF-based RZSM estimates and in situ RZSM measurements were disregarded. As a result, a total of 1509 in situ sites were considered. Note that the EF model structural uncertainty computed at the point scale is assumed to be representative for the coarse scale as well.

Finally, the EF structural uncertainties obtained from Eq. (8) add to the propagated RZSM uncertainty budget (Eq. (4)) as:

$$\sigma(RZSM_n) = \sqrt{\Delta_n^2 + \left(\frac{\partial RZSM_n}{\partial T}\right)^2 \sigma^2(T) + \sigma^2(EF)} \tag{9}$$

## 4 Results and discussion

In this section, we first show results of the $T$-parameter optimization. Next, we compare the RZSM product globally to E5L. We then discuss the estimates for EF model structural uncertainties. Finally, we compare our RZSM uncertainty estimates with those obtained with the baseline method.

### 4.1 $T$-parameter optimization

After filtering out unreliable data (see Section 2.2), 3901 ISMN time series from 67 different measuring depths between 0 and 200 cm were available for the $T$-optimization process. Figure 1 shows the distribution of $T_{opt}$ values binned into our four chosen RZSM layers (0–10 cm, 10–40 cm, 40–100 cm, and 100–200 cm). The median $T_{opt}$ values for these layers were 6, 15, 48 and 70 days, increasing with soil depth as expected (Paulik et al., 2014; Wang et al., 2017). These median $T_{opt}$ values were
then used to compute RZSM globally.





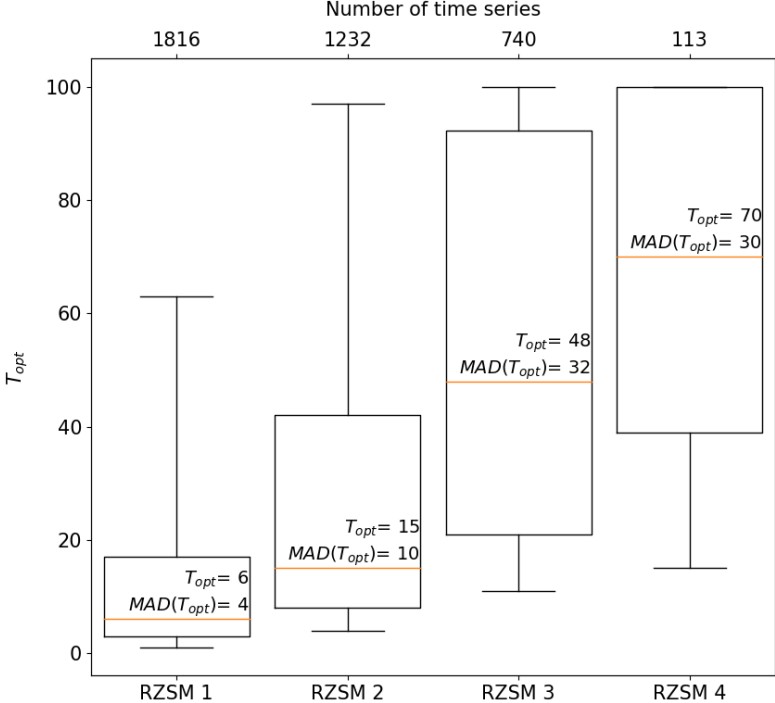

**Figure 1.** $T_{opt}$ values calibrated on 3901 in situ time series and binned per RZSM layers 1–4. Median values (represented by orange lines) from each bin were used to compute a global RZSM product. Median absolute deviations ($MAD(T_{opt})$) were used in estimating RZSM uncertainties.

A five-fold cross-validation was performed to verify the robustness of this approach. The variability in median $T_{opt}$ values per soil layer is increasing with depth but remains negligibly small in all layers, with 6, 15–16, 47–50 and 67–72 for soil layers 1–4, respectively (Figure 2a). Subsequently, the five median $T_{opt}$ values derived from the training subsets were used to estimate RZSM for the different layers of the respective validation sets (Figure 2c) and resulted in Pearson's r of 0.64–0.67,

0.64–0.65, 0.57–0.6, and 0.48–0.6 for soil layers 1–4, respectively. When evaluating each training set directly, correlations were 0.65–0.66, 0.65, 0.58–0.59, and 0.53–0.56 for soil layers 1–4, respectively (Figure 2b).





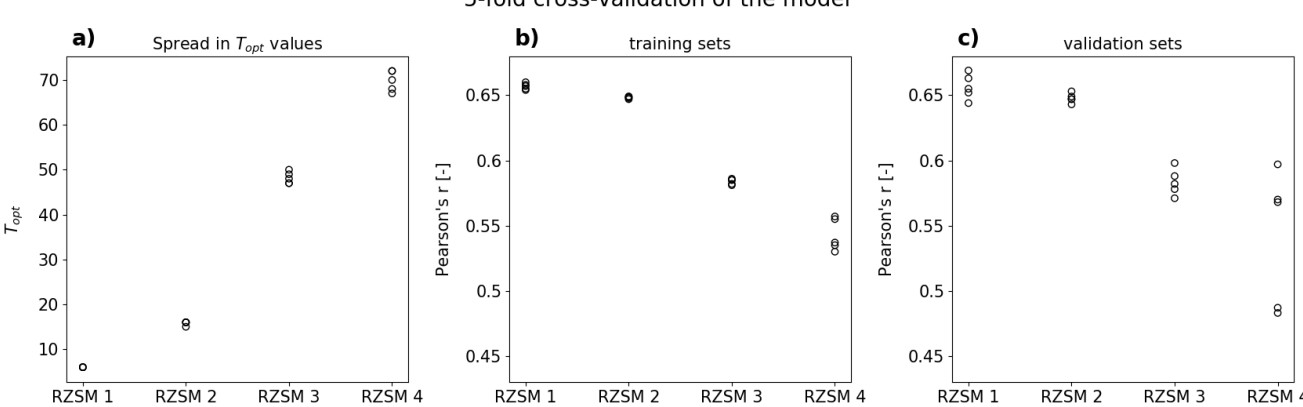

**Figure 2.** Cross validation results showing the spread in $T_{opt}$ values (a), and agreement of the training (b) and validation (c) sets with in situ data.

The little variability between the validation and training sets suggests that $T_{opt}$ values are not over-fitted to ISMN site conditions and can be used robustly in other regions as well. Notably, the spread in median $T_{opt}$ values increases with soil depth while the correlation scores decrease. This indicates reduced reliability of the method in deeper soil layers, which is in line with the assumption that the coupling between the surface and root-zone SM decreases with depth. Note, however, that results for deeper layers are also affected by the smaller sample sizes at greater depths.

### 4.2 Global RZSM product quality assessment

A global SM dataset spanning the 2002–2020 period was computed using the EF method and $T$-parameters optimized with the approach described in section 4.1. Figure 3 shows a comparison between all of the RZSM product layers as well as the input C3S SSM dataset with E5L.

RZSM product layers agree best with the (approximately) matching E5L depth layers in all but one case. The highest median Pearson correlations with the E5L reference layers (0–7 cm, 7–28 cm, 28–100 cm, and 100–289 cm) were obtained by C3S SSM (0.55), RZSM layer 1 (0.49), RZSM layer 3 (0.41), and RZSM layer 4 (0.28), respectively. Even though C3S SSM correlates best with the E5L surface layer, the correlation score for RZSM layer 1 (0–10 cm) is only insignificantly smaller (0.54). Similarly, the second E5L layer (7–28 cm) best agrees with RZSM layer 1 (r=0.49) but the layer most congruent in depth (RZSM 2) is a close second (r=0.47). In the remaining depth layers, correlations between C3S SSM and E5L are substantially lower than for the RZSM product, which proves the ability of our EF approach to approximate SM below the surface layer. While RZSM layer 1 (0–10 cm) shows the best agreement with E5L layer 2 (7–28 cm; r=0.49) RZSM layer 2 (10–40 cm) correlates only slightly less with that layer (r=0.47).

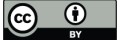



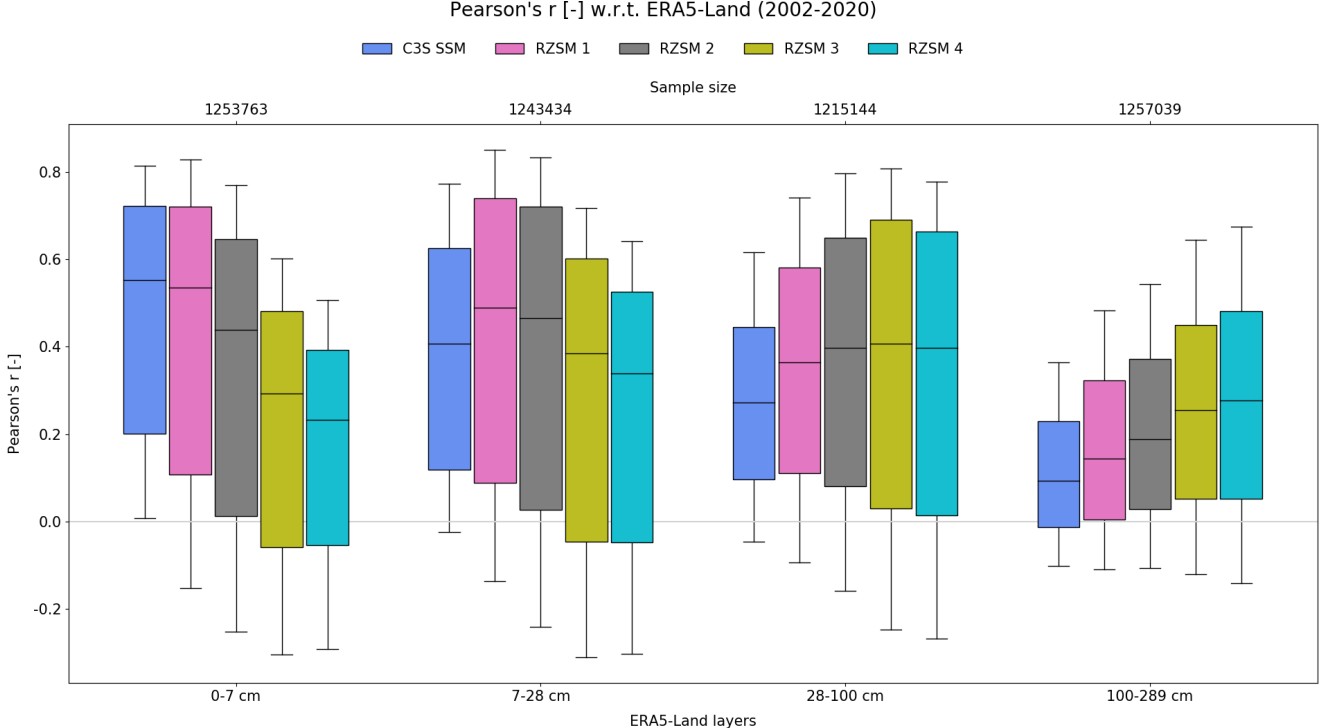

**Figure 3.** Product intercomparison of the C3S SSM and RZSM products against E5L SM.

Results are also consistent with the assumption of the EF model that SM dynamics decrease with depth and that $T_{opt}$ ought to increase accordingly, as was also found by other studies (Wagner et al., 1999; Paulik et al., 2014; Wang et al., 2017; Beck et al., 2009; Mahmood and Hubbard, 2007). At the same time, the maximum correlation values decrease with depth confirming the diminishing coupling between the surface and root-zone layers, as also found at in situ station level 2 and demonstrated by others (Paulik et al., 2014; Brocca et al., 2010; Sure and Dikshit, 2019).

## 4.3 EF model structural uncertainty

Figure 4 shows estimates for the model structural uncertainties (Section 3.2.3) obtained at all available in situ sites, binned into the four RZSM product layers. Their median values (represented by orange lines and annotated) were used as estimates for $\sigma(EF)$. Note that in situ measurement errors were assumed to be negligible and thus not influencing ubRMSD estimates, which likely causes model structural uncertainties to be overestimated. Also, structural uncertainties are assumed to be constant in time.





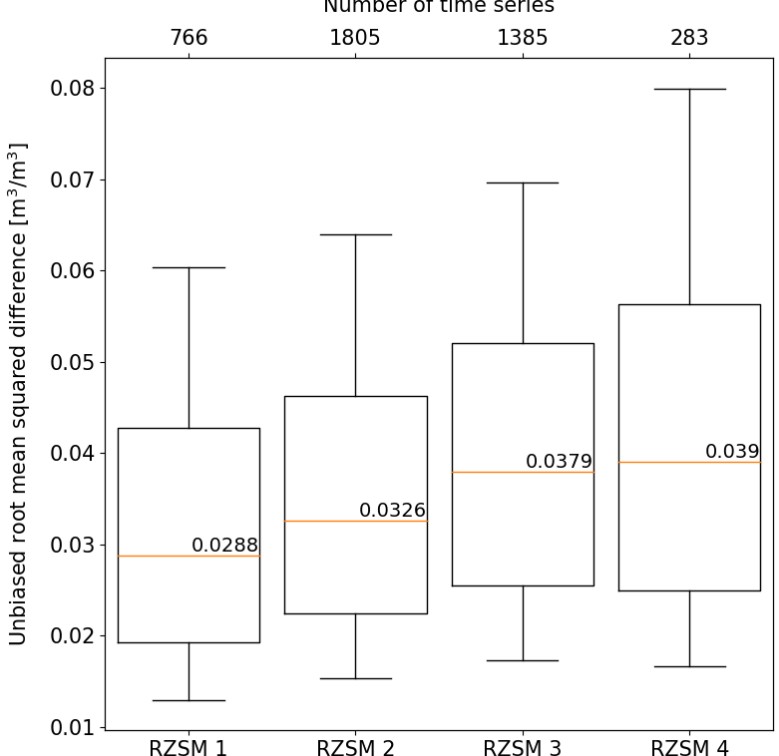

**Figure 4.** The ubRMSD between propagated RZSM from in situ SSM using the EF model and measurements of RZSM at the same location and in the same depth, calculated at 1509 different sites. The median ubRMSD value for each bin (represented by orange lines and annotated) represents $\sigma(EF)$ for the according $T_{opt}$.

As anticipated, an increase in $\sigma(EF)$ corresponds to the growing distance between the surface and the root-zone measurements, demonstrating the decreasing coupling strength between both layers. Note that $\sigma(EF)$ shows significant variability within RZSM layers which is likely, at least to some degree, related to variations in local conditions. However, as with the $T$-parameter optimization, we estimate structural uncertainties based only on a limited number of in situ stations and, therefore, use the median to extrapolate globally.

## 4.4 RZSM uncertainty budget calculation

Figure 5 compares RZSM uncertainty estimates obtained from the baseline method (De Santis and Biondi, 2018) with those from the approach proposed here. Figure 5a shows a time series of RZSM uncertainties from the baseline method at an arbitrary location. Figure 5b shows the effect of changing $\sigma(T)$ from 10% of $T_{opt}$ to the median absolute deviation of $T_{opt}$, which is an amplified temporal variability. Simultaneously, mean uncertainty values increase and get closer to the magnitudes of the input SSM dataset. Moreover, they no longer diminish with increasing $T$-values (i.e., depth) as is the case in the baseline formulation.




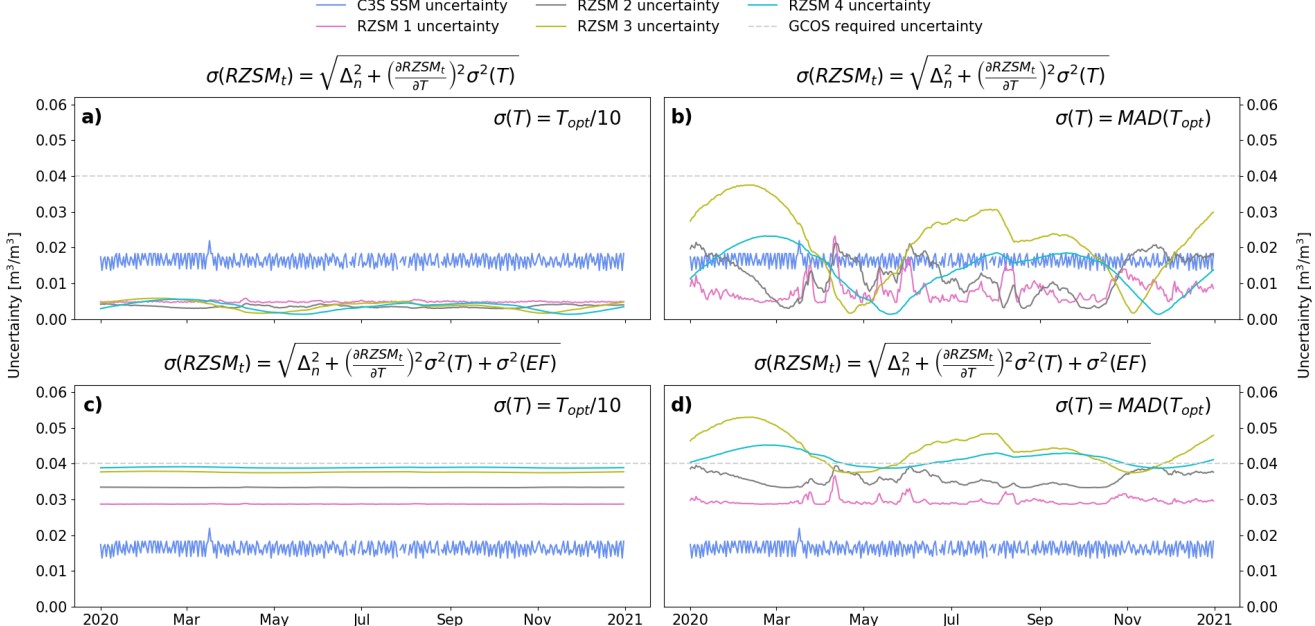

**Figure 5.** Evaluation of the impact of changes to the baseline method illustrated on an example 2020 time series from 9.875N, 1,625E. C3S SSM uncertainties were propagated with the baseline scheme in a), while b) and c) show the individual impacts of increasing the noise of $T$ from 10% of $T_{opt}$ to $MAD(T_{opt})$ and adding the term $\sigma^2(EF)$, respectively. Combined effects of both changes are shown in d). The dashed grey line indicates target RZSM uncertainty level as required by GCOS (2022).

This is presumably more realistic since the progressive decoupling between the surface and deeper soil layers can be expected to cause uncertainties to increase rather than to decrease.

Figure 5c shows the impact of accounting for $\sigma(EF)$ in the total uncertainty budget when using 10% of $T_{opt}$ as $T$-parameter uncertainty ($\sigma(T)$). Considering this term substantially increases the magnitude of the propagated uncertainties and leads them to increase with depth (as does $\sigma(EF)$). However, the uncertainties' temporal variability is reduced substantially as the effect of $\sigma(T)$ is overshadowed by that of $\sigma(EF)$. Finally, Figure 5d shows the combined effect of using the MAD of $T_{opt}$ as its parameter noise $\sigma(T)$ and accounting for model structural uncertainty $\sigma(EF)$. Compared to the baseline (Figure 5a), this yields an increased overall magnitude of the uncertainties, a more realistic increase in (temporal average) uncertainties with depth, and an amplified temporal variability in all layers during transitions between dry and wet conditions, which is also expected (see Figure 6).

## 4.5 Assessment of uncertainty estimates

Similar as in De Santis and Biondi (2018), we assess the use of the proposed MAD estimates for $\sigma(T)$ by computing Pearson's $r$ and root-mean-square differences (RMSD) with respect to in situ data before and after removing a fixed percentage of the data



(5, 10, 15, and 20%) with highest uncertainty estimates. In case of effective correspondence between high values of both the estimated RZSM uncertainties and the observed RZSM deviations from reference in situ measurements, it is expected that the skill metrics will improve due to the masking. This hypothesized correspondence holds well as long as the difference between in situ and satellite-based RZSM values is mainly due to the random errors of the latter. Note that this analysis is only sensitive to the impact of using different values for $\sigma^2(T)$ (($T_{opt}/10)^2$ versus $MAD(T_{opt})^2$) since the estimated structural uncertainty

$\sigma(EF)$ is constant in time and therefore cannot change the ranking of the total uncertainties.

Figure 6 shows both SM and uncertainty estimates for C3S SSM and RZSM layer 2 estimates for the same site as in Figure 5 (9.875N, 1,625E) and highlighting 20% of data with the highest uncertainties. Overall, despite the differences in magnitude and amplitude, both our and the baseline method assign the highest uncertainty values to timestamps corresponding to significant soil wetting or drying events. However, in the baseline method the average magnitude of SSM input uncertainty appears to

have a greater influence on the calculated RZSM uncertainty estimates. This is most evident when comparing values before and after the inclusion of Metop-A ASCAT into the C3S product in January 2007 (indicated by the dashed vertical line in Figure 6), which substantially improved data quality thereafter. Specifically, mean C3S SSM uncertainty dropped from 0.029 $m^3/m^3$ to 0.018 $m^3/m^3$. Such a clear shift is also visible in the uncertainty values propagated with the baseline method (from 0.008 $m^3/m^3$ before to 0.004 $m^3/m^3$ after the introduction of Metop-A ASCAT). This causes the baseline method to predict

the majority of the 20% most uncertain SM values to occur in the pre-ASCAT period. In contrast, in our approach, average uncertainties remain stable (at 0.036 $m^3/m^3$) over the entire time period. This suggests that the use of $MAD(T_{opt})^2$ as an estimate for $T$-parameter uncertainty reduces the sensitivity to structural breaks, i.e., large variations between the uncertainties of the C3S SSM input sensors, and improves the method's capability to predict day-to-day uncertainty variations. Lastly, after the introduction of ASCAT, both schemes consistently assign higher uncertainties to timestamps characterized by large SM

changes. Taken together, while the use of $T_{opt}/10$ as $T$ parameter uncertainty seems to yield realistic estimates for uncertainty variations due to the use of different C3S SSM input sensors, using $MAD(T_{opt})$ as $T$ parameter uncertainty seems to better predict day-to-day uncertainty variations in the RZSM estimates.

Figure 7 shows the results of the data removal experiment described above, summarized for all considered ISMN stations. To compare the performance with and without the effect of C3S structural breaks on the uncertainty values (see above), results are

shown for both the full product period (2002–2020; Figure 7a–d) and a sub-period without breaks, i.e., from the inclusion of SMAP data onward (April 1st 2015–2020; Figure 7e–h). In both cases, correlation coefficients obtained for the complete time series were compared to those obtained after removing 5, 10, 15, and 20 % of data with the highest associated uncertainties.

In case of the full product period (Figure 7a–d), using $\sigma(T) = T_{opt}/10$ as $T$ parameter uncertainty seems to consistently yield more realistic estimates of temporal uncertainty variations than using $\sigma(T) = MAD(T_{opt})$. This is true for all four soil

layers. Masking out more uncertain data indicated by either method consistently improves agreement with in situ reference data in the first two product layers. This improvement increases the more data are masked out, as is expected. In the absence of such breaks (Figure 7e–h) RZSM uncertainty variations seem to be better predicted when using $\sigma(T) = MAD(T_{opt})$ as $T$ parameter uncertainty in almost all cases. Notably, in layers 3 and 4, data removal according to either method degraded the agreement with field measurements.



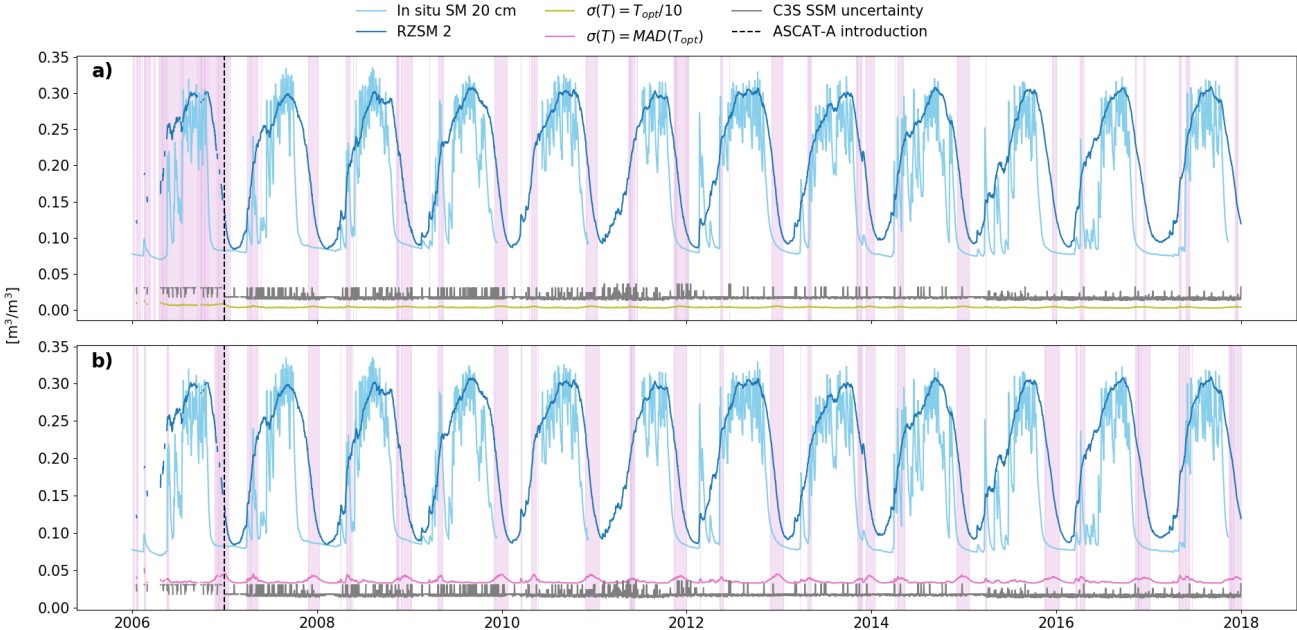

**Figure 6.** Differences in uncertainty variations of the baseline (a) and our proposed uncertainty estimation approach (b). Illustrated on the example of AMMA-CATCH: Belefoungou-Mid (9.875N, 1,625E) field measurements from 20 cm depth.

In summary, the propagation of C3S SSM input uncertainties yields accurate predictions of temporal uncertainty variations of RZSM estimates obtained with the EF method for the first two layers (0–10 cm and 10–40 cm). This is no longer the case for deeper layers (40–100 cm and 100-200 cm). Note, however, that the RZSM estimates in these layers themselves still exhibit reasonable skill when evaluated against E5L (see Figure 3).

## 5 Summary and Conclusions

In this study, we computed root-zone soil moisture (RZSM) globally in four depth layers (0–10 cm, 10–40 cm, 40–100 cm, and 100-200 cm) from merged satellite surface soil moisture (SSM) retrievals of the Copernicus Climate Change Service (C3S) COMBINED product v202012 using the exponential filter (EF) method. The EF model parameter $T$ has been optimized at point scale by maximizing the correlation against globally-distributed in situ SM measurements from the International Soil Moisture Network (ISMN). The median of the optimized $T$ values at each layer have been used to compute the global product. A global product intercomparison with ERA5-Land (E5L) reanalysis SM data has shown a satisfactory level of agreement in all layers (global median correlations of the four above-mentioned product layers against E5L reference layers 0–7 cm, 7–28 cm, 28-100 cm, and 100-289 cm were 0.54, 0.47, 0.41, and 0.28, respectively).

Uncertainties in the RZSM estimates obtained with the EF method were calculated using the law for the propagation of uncertainties. Uncertainties of the input SSM data were available in the C3S product and have been calculated by the data





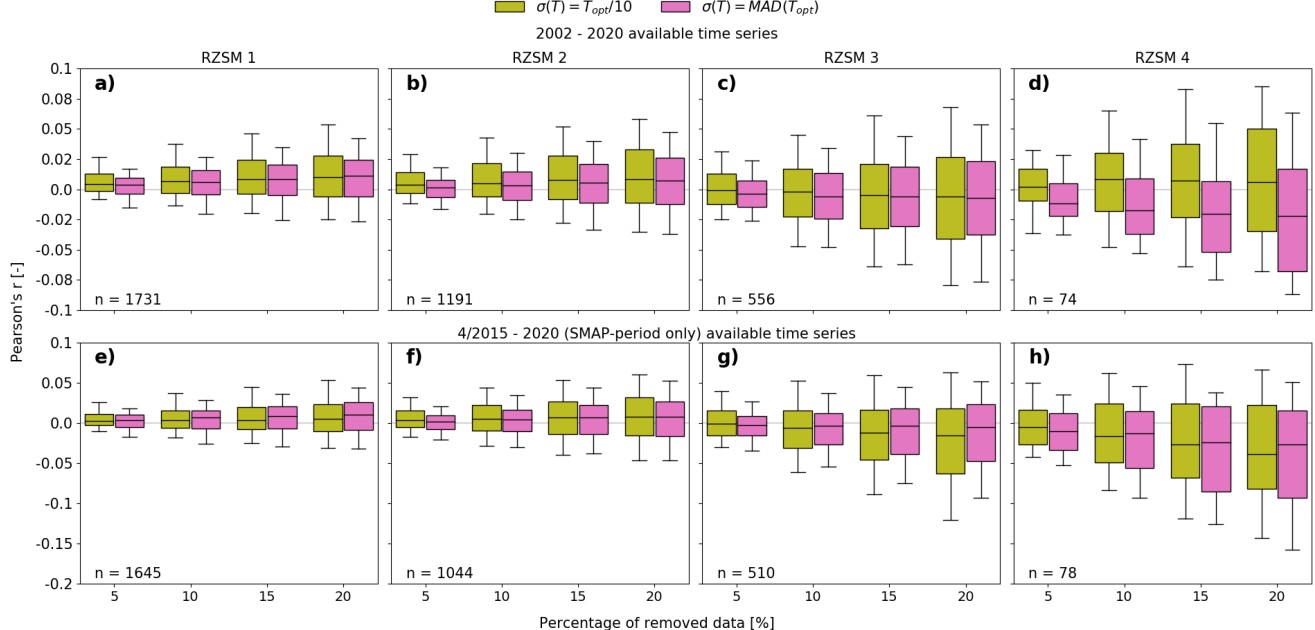

**Figure 7.** Correlations with in situ measurements (y-axis) before and after removing a fixed percentage of data with the highest uncertainty (x-axis) for the period 2002–2020 (a–d) and 2015–2020 (e–h). Uncertainties were calculated using either $\sigma(T) = T_{opt}/10$ (olive colors) or $\sigma(T) = MAD(T_{opt})$ (orchid colors).

producers using Triple Collocation Analysis (TCA). We tested the use of the median absolute deviation of optimized $T$ parameters at the available ISMN locations ($MAD(T_{opt})$) as a proxy for $T$ parameter noise. Results obtained using $MAD(T_{opt})$ in uncertainty propagation were compared with results obtained using 10% of the optimized $T$ parameter itself ($T_{opt}/10$) used in earlier studies. While the use of $T_{opt}/10$ as $T$ parameter uncertainty seems to yield realistic estimates for uncertainty variations due to the use of different C3S SSM input sensors, using $MAD(T_{opt})$ as $T$ parameter uncertainty seems to better predict day-to-day uncertainty variations in the RZSM estimates.

Even though propagating SSM input and model parameter uncertainties yields credible predictions of temporal uncertainty variations, absolute uncertainty magnitudes appear unrealistically small (below 0.01 $m^3/m^3$). This is because the propagation of uncertainty only accounts for uncertainties in the data and parameters input to the EF method, but not for limitations of the EF method itself (e.g., the progressive inability of the method to model deeper-layer RZSM due to vanishing surface–root zone coupling). We proposed to estimate these EF model structural uncertainties as the unbiased root-mean-square differences between RZSM estimates for each of our four product depth layers obtained by applying the EF method to in situ SSM measurements, and actual in situ RZSM measurements taken at the same location and depth. This was done at all available ISMN sites and the median of these estimates used as a global proxy of EF structural uncertainty for each of the four product depth layers, respectively. Combined, propagated SSM input and model parameter uncertainties and EF structural uncertainties were



considered to yield realistic estimates of the total RZSM product uncertainty budget in all layers (global mean uncertainties of the four product layers are 0.031, 0.035, 0.04, and 0.04). Note, however, that a quantitative validation of uncertainty magnitudes is still pending due to the lack of reliable uncertainty reference data on a global scale and for different RZSM depth layers.

The EF parameter uncertainty was estimated on a global scale and can be expected to differ for smaller scales, especially where the variability in environmental conditions is lower. Similarly, estimates of the EF model structural uncertainty are likely

to differ on local to regional scales. Also, the structural uncertainty of the EF, here assumed to be constant in time, could in fact vary on a sub-seasonal scale given the phenomena that regulate the process of water transfer in the soil. Moreover, random errors of the in situ measurements were assumed to be negligible and were not accounted for in estimating the structural uncertainty of the model. Nonetheless, it is plausible that the EF structural uncertainty is much greater than the random uncertainty of the in situ sensors. Estimates of the random uncertainty of in situ sensors could allow for a more accurate estimation of the EF

structural uncertainty in the future.

Further insights could also be gained by evaluating the behaviour of the proposed method in propagating uncertainties of different SSM input data, i.e., single-sensor products without structural breaks and non-static input SSM uncertainties obtained by different means than TCA. Nonetheless, this study is an important step towards understanding and describing the uncertainties of EF-based RZSM products.

**6  Code and data availability**

Python package used in the computation of the root-zone soil moisture data and its associated uncertainties from surface soil moisture observations by means of exponential filter: https://github.com/TUW-GEO/pyswi

Global root-zone soil moisture data produced and utilized in this study, available for 2002-2020 period as daily image files in netCDF4 format: https://doi.org/10.48436/9gsg6-nn854





**Appendix A: ISMN references**

Table A1: ISMN networks used in this study.

| Network | Time series used for $T$-parameter optimization | Time series used for EF model structural uncertainty estimation | Reference |
|---|---|---|---|
| AMMA-CATCH | 31 | 27 | Mougin et al. (2009); Cappelaere et al. (2009); de Rosnay et al. (2009); Lebel et al. (2009); Galle et al. (2015) |
| ARM | 90 | 113 | Cook (2016a, b, 2018) |
| AWDN | 112 | 148 | - |
| BIEBRZA_S-1 | 15 | 18 | Musial et al. (2016) |
| BNZ-LTER | 7 | 22 | Van Cleve et al. (2015) |
| CALABRIA | 12 | - | Brocca et al. (2011) |
| CAMPANIA | 1 | - | Brocca et al. (2011) |
| COSMOS | 65 | 2 | Zreda et al. (2008, 2012) |
| CTP-SMTMN | 147 | 167 | Yang et al. (2013) |
| DAHRA | 4 | 4 | Tagesson et al. (2015) |
| FLUXNET-AMERIFLUX | 22 | 16 | - |
| FMI | 16 | 37 | Ikonen et al. (2016, 2018) |
| FR_Aqui | 28 | 23 | Al-Yaari et al. (2018); Wigneron et al. (2018) |
| GROW | 118 | - | Xaver et al. (2020); Zappa et al. (2019, 2020) |
| GTK | - | 24 | - |
| HiWATER_EHWSN | - | 1 | Kang et al. (2014); Jin et al. (2014) |
| HOAL | 90 | 97 | Blöschl et al. (2016); Vreugdenhil et al. (2013) |
| HOBE | 64 | 60 | Jensen and Refsgaard (2018); Bircher et al. (2012) |
| HSC_SEOLMACHEON | 1 | - | - |



| | | | |
|---|---|---|---|
| HYDROL-NET_PERUGIA | 4 | 6 | Flammini et al. (2018a, b); Morbidelli et al. (2011, 2014, 2017) |
| ICN | - | 24 | Hollinger and Isard (1994) |
| IIT_KANPUR | - | 3 | - |
| IMA_CAN1 | 9 | - | Biddoccu et al. (2016); Raffelli et al. (2017); Capello et al. (2019) |
| IPE | 1 | - | Alday et al. (2020) |
| iRON | 5 | 16 | Osenga et al. (2019, 2021) |
| KIHS_CMC | 54 | 38 | - |
| KIHS_SMC | 51 | 32 | - |
| LAB-net | 2 | 1 | Mattar et al. (2014, 2016) |
| MAQU | 53 | 62 | Su et al. (2011); Dente et al. (2012) |
| MOL-RAO | 9 | 10 | Beyrich and Adam (2007) |
| MySMNet | 15 | 11 | Kang et al. (2019) |
| NAQU | 5 | 31 | Su et al. (2011); Dente et al. (2012) |
| NGARI | 5 | 84 | Su et al. (2011); Dente et al. (2012) |
| NVE | - | 10 | - |
| ORACLE | 24 | 32 | - |
| OZNET | 101 | 105 | Young et al. (2008); Smith et al. (2012) |
| PBO_H2O | 115 | - | Larson et al. (2008) |
| PTSMN | 80 | 60 | Hajdu et al. (2019) |
| REMEDHUS | 22 | - | González-Zamora et al. (2019) |
| RISMA | 51 | 62 | Canisius (2011); L'Heureux (2011); Ojo et al. (2015) |
| RSMN | 13 | - | - |
| SASMAS | 27 | 13 | Rüdiger et al. (2007) |
| SCAN | 575 | 806 | Schaefer et al. (2007) |
| SKKU | 56 | 42 | Nguyen et al. (2017) |
| SMN-SDR | 76 | 127 | Zhao et al. (2020); Zheng et al. (2022) |
| SMOSMANIA | 79 | 66 | Calvet et al. (2007); Albergel et al. (2008); Calvet et al. (2016) |
| SNOTEL | 788 | 942 | Leavesley et al. (2008) |



| | | | |
|---|---|---|---|
| SOILSCAPE | 385 | 247 | Moghaddam et al. (2010, 2016); Shuman et al. (2010) |
| SWEX_POLAND | 6 | 17 | Marczewski et al. (2010) |
| TAHMO | 68 | 10 | - |
| TERENO | 14 | 10 | Zacharias et al. (2011); Bogena et al. (2012, 2018); Bogena (2016) |
| UDC_SMOS | 16 | 11 | Loew et al. (2009); Schlenz et al. (2012) |
| UMBRIA | 37 | 28 | Brocca et al. (2008, 2009, 2011) |
| UMSUOL | 4 | 6 | - |
| USCRN | 309 | 358 | Bell et al. (2013) |
| USDA-ARS | 4 | - | Jackson et al. (2010) |
| VAS | 1 | - | - |
| VDS | 12 | 8 | - |
| WEGENERNET | 1 | - | Kirchengast et al. (2014); Fuchsberger et al. (2021) |
| WSMN | 1 | - | Petropoulos and McCalmont (2017) |



**Figure A1.** Location map of the ISMN in situ stations used in this study and listed in TableA1.

*Author contributions.* Adam Pasik: Conceptualization, Formal analysis, Investigation, Visualization, Software, Writing - original draft preparation. Alexander Gruber: Conceptualization, Methodology, Supervision, Writing - review & editing. Wolfgang Preimesberger: Data curation, Software, Validation, Writing - review & editing. Domenico De Santis: Methodology, Software, Writing - review & editing. Wouter Dorigo: Conceptualization, Methodology, Funding acquisition, Supervision, Writing - review & editing.


*Competing interests.* The authors declare that they have no conflict of interest.

*Acknowledgements.* This study received funding from the European Union's Horizon 2020 research and innovation programme under grant agreement nº 870353 - Global Gravity-based Groundwater Product (G3P) project. G3P is funded in response to the Earth observation call



LC-SPACE-04-EO-2019-2020 "Copernicus evolution – Research activities in support of cross-cutting applications between Copernicus
services", as part of the H2020-SPACE-2018-2020 activity "Leadership in Industrial Technologies - Space Part". Please visit https://g3p.eu
for further information on the project.



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
