# Peer review of "Uncertainty estimation for a new exponential filter-based long-term root-zone soil moisture dataset from C3S surface observations"

_EGUsphere, 2023_

## Author Comment (AC1)

**Responses to reviewer's comments**

We thank the editor and the reviewers for their time and effort to review our manuscript, which helped to further increase the quality of the paper. All comments have been addressed carefully.

Below, reviewer comments are marked in red.
Responses to the comments are marked in blue.
Cited changes that have been made in the manuscript are marked in *italic*.

**Reviewer 1:**

The MS titled, 'Uncertanity estimation for a new exponential filter-based long-term root-zone soil moisture dataset from C3S surface observations' by Pasik et al. describes a methodology to deduce rootzone SM using only satellite derived surface soil moisture using a well-known exponential filter approach with its error characterization. In general, I found this study lacking in novelty factor. The exponential model has been around for a while now and is shown to have limited success in estimating SM at deeper layers (>40cm) as also shown in this study. Furthermore, there are some glaring gaps in background infromation, method descriptions etc. Therefore, despite tackling a critical issue in sub-surface hydrology, I would recommend rejection in its current form. However, I would be more than happy to see it re-submitted with significant revisions.

We regret to learn that the novelty in the uncertainty estimation scheme for the exponential filter (EF) is not clear. Our intention is for the uncertainty estimation method (rather than the dataset or the EF itself) to be the main focus of the paper. To our best knowledge, this study is the first one to provide uncertainty estimates for a global, EF-based soil moisture dataset which also accounts for uncertainties of the EF method itself. Moreover, while the application of the EF approach at greater soil depths yields less accurate results, we believe that there can still be merit found in such estimates provided that the uncertainties are known (which, again, are the main focus of this study). We will revise the manuscript to make this more clear, and address the gaps in background information and method descriptions mentioned in the major and specific comments below.

| Major Comments | Reply |
|---|---|
| 1) As mentioned earlier, the exponential filters have been studied extensively (as also acknowledged in the MS) in the past. Also has well docmented issues such as poor performance in the deeper layers, struggles with widely different soil types between surface and lower layers (no mention or discussion around this in the MS), summer season decoupling etc. This study acknowledges and re-affirms most of these issues but presents no path forward in trying to solve them. Its does not seem to be taking the scinece forward either by reducing the model known limitations, or sheding new lights on model performance with its global implementation or discussion. Therefore, in my opition, in the current form, this study lack the novelty factor and may need significant revisions. | Thank you for pointing out these previously unacknowledged limitations of the EF. We will mention them while discussing the limitations of the method: *"Other limitations of the method include generally poorer performance in arid zones and when soil texture is not homogeneous throughout the soil column (Yang et al., 2022; Ford et al. 2014)"*. In this MS, we focus on advancing the understanding of the EF method by contributing our uncertainty estimation scheme to describe, rather than reduce, the model's known limitations. We will make this intention clear by adding the following sentence to conclude section **3.1 Exponential filter**: *"It is precisely such limitations that we attempt to describe with the uncertainty estimation scheme developed in this study, and hence advance the understanding of the EF method's performance"*. |
| 2) I agree with the authors that their product could be the covering the longest period of record for observation-based RZSM, but shoud have at least acknowledged other global products such as SMAP and SMOS L4 products in the MS. In fact, SMOS L4 product is similar to this study, they are also using EF (https://sextant.ifremer.fr/record/316e77af-cb72-4312-96a3-3011cc5068d4/) whereas SMAP L4 uses data assimialtion approach to merge SMAP with the Catchment Land surface Model with detailed uncertainty analysis and published ATBDs. | Thank you for the suggestion. We will explicitly reference SMAP L4 RZSM while discussing various approaches to estimating RZSM in the **Introduction**: *"Satellite-based SSM observations can also be assimilated into a land surface model to produce estimates of RZSM with global coverage, as in the case of the SMAP L4 RZSM product (Reichle et al., 2017)."* We will also edit the concluding sentences of the introduction to acknowledge other EF-based datasets (incl. SMOS L4 RZSM): *"While other EF-based datasets exist (e.g., the SMOS L4 product), they offer limited spatio-temporal coverage and lack quantitative uncertainty information (Al Bitar and Mahmoodi, 2020; Bauer-Marschallinger et al., 2018)."* |

3) Furthertmore, I think the MS could be improved further with some more background information on other rootzone SM estimation techniques, currently the MS does not talk about other methods and why EF might be better than others, in that context an best average correlation of 0.56 doesnt not inspire too much confidence. The authors simply skipped over some of the detailes of satellite SM estiamtions like difference in bands (X&C for AMSR-E vs L for SMAP and SMAOS etc.), I think it would improve MS.

In addition to the discussion on the root-zone estimation techniques (and the EF method in their context) already present in L44-60 of the submitted MS, we will also include specific references to SMOS L4 and SMAP L4 RZSM products as per the previous comment. Furthermore, in the context of product performance, we will add the following as the concluding sentence of section **4.2 Global RZSM product quality assessment**: *"The performance of our product is similar to that of other satellite-based RZSM products found in other studies, especially when considering the same regions for assessment (Xu et al. 2021, Reichle et al. 2017). While the data set presented here does not outperform other existing RZSM products, it distinguishes itself as the only purely observation-based global product covering such a long time period, and the only EF-based product that has uncertainty estimates provided with it"*.

The input C3S SM dataset is a harmonized product where the biases between bands are mitigated by the inter-calibration of the sensors. Furthermore, each estimate is a weighted combination of the individual sensor observations available on that day. However, the spectral band of the sensor, via its suitability for retrieving soil moisture information (e.g., different sensitivity to precipitation and evaporation), impacts the uncertainty estimates. We will make this more clear when introducing the C3S dataset in **section 2.1** by including the following: *"Note that the distinctive life spans and spectral bands of the used satellite missions (e.g., C and X-bands used by AMSR-E, L-band used by SMOS and SMAP) can potentially also lead to distinctive changes in the data quality of the merged product via the differences in their sensitivity to precipitation or evaporation. These sudden changes in SSM and uncertainty data are hereinafter referred to as systemic breaks (Preimesberger et al., 2021). Although said breaks have a marginal impact on the SSM signal itself due to the inter-calibration of sensors, they are distinct in the uncertainty estimates. As more and newer sensors provide better retrievals, mean uncertainty values typically decrease distinctively with every new satellite launch in more recent periods (Gruber et al., 2017)"*.

| | |
|---|---|
| 4) If I understand correctly, most of the in-situ sites are in open fields (usually near agricultural land). Therefore, Topt obtained may only be able to represent (presumabely) a particular landcover type. Is there any analysis authors have performed to assess the model perfromance at other locations? While reading the MS, I could not figure out if the model was implemented at gridded scale or only at the ISMN sites. Perhaps, this could be made more clearer. | It is correctly pointed out that the reference in situ sites used for optimizing the T-parameter do not equally represent the variety of land cover classes. Similarly, their geographical distribution is largely skewed towards the Global North (as acknowledged in L27-29 of the submitted MS). However, as other studies have shown, optimizing the T-parameter per, e.g., soil type, does not yield an improvement versus using an averaged T-value (e.g., De Lange et al. 2008, Grillakis et al. 2021). This is why we choose to lump all the available in situ stations together in the optimization of the T-parameter. Even though their distribution might be skewed toward particular land cover classes (i.e., grasslands or croplands) the impact of this is probably negligible (e.g., Stefan et al. 2021). This limited sensitivity to variation in T-parameter values is discussed in L135-141 of the submitted MS.

Indeed we evaluate our product globally at grid cell level against ERA5-Land data. We will include spatial correlation maps for each of the product layers in an additional new Figure (please find it included at the end of this document), where the performance of the RZSM product can be seen at every single location. Performance patterns observed in these maps do not show correlation with the distribution of the ISMN stations used for the T-parameter optimization (i.e., Fig. A1 in the submitted MS), which too suggests that the impact of their uneven distribution and over-representation of certain land cover classes does not have a substantial impact.

The EF model was optimized with point-scale ground measurements and implemented globally at the grid scale. We will add the following text to make this clearer throughout the MS:

1) in the opening section of the **Results**: *"In this section, we first show results of the point-scale T-parameter optimization. Next, we compare the gridded RZSM product globally to E5L."*

2) in the opening sentence of section **4.2 Global RZSM product quality assessment**: *"A global SM dataset spanning the 2002–2020 period was computed using the EF method and T-parameters optimized at the point scale with the approach described in section 4.1."* |

5) Finally, I think the discussoin could be further improved (especially if study is simply focused in implementing at larger scale with its limitations intact) by talking about if there is any regional pattern in model performance (arid vs humid conditions); tropical vs sub-tropical region? Does soil types play any role in Topt and uncertainty estimations? What is the dominent landcover type and can these different rooting systems (barren soils vs cropland vs deep rooted trees) explain some of the issues beign faced?

Thank you for this suggestion, we will include spatial correlation maps in a new figure (mentioned in our response to comment #4) and discuss the observed patterns in section **4.2 Global RZSM product quality assessment**: *"A global SM dataset spanning the 2002–2020 period was computed using the EF method and T-parameters optimized at the point scale with the approach described in section 4.1. Figures Xa–e) show correlation maps of each of the RZSM product layers as well as the input C3S SSM dataset with E5L. The spatial patterns observed in the C3S SSM data (Figure Xa) are strikingly similar to those in RZSM layer 1 (Figure Xb) with slight to moderate deterioration in performance over the high latitudes ($> 60°N$). This is not surprising given that both products differ only by a small degree of smoothing applied to RZSM layer 1 and are compared to the same E5L layer (0-7 cm). RZSM layers 2 and 3 (Figure Xc-d) are compared to E5L layers 7-28 and 28-100 cm, respectively, and largely preserve good performance in regions where the input C3S SSM product also performs well, i.e., in Europe (bar Scandinavia), the Caspian and Aral Sea basins, the Eastern United States, India, Southeast Asia, South America, Sub-Saharan Africa, and Australia. At the same time, deterioration of performance is observed in high latitudes and in arid environments such as the Sahara desert and the Arabian Peninsula where the reduced strength of coupling between the surface and root-zone dynamics may hinder the EF performance (Yang et al., 2022). The patterns of good and poor performance visible in RZSM layers 1-3, are not replicated in RZSM layer 4 (Figure Xe) where the agreement with the reference E5L 100-289 cm layer is spatially very heterogeneous and worse overall. The few regions where the good performance observed in shallower layers is preserved include India, Southeast Asia, and the Eastern United States"*.

The impact of soil type or land cover on the performance of the EF remains ambiguous (e,g., Stefan et al. 2021). External variables could potentially influence the uncertainty estimates, but such detailed analysis is outside of the scope of this study. We will reiterate the focus of this study throughout the MS, e.g., as per major comment #1 and also by adding the following concluding sentences to the MS' Introduction: *"The focus and novelty of this paper lie in quantifying, rather than reducing, the EF model's known limitations by providing a methodology for comprehensive uncertainty estimation for the EF method. Additionally, to our best knowledge, this dataset is, as yet, the longest available solely observation-based, error-*

| Specific Comments | Reply |
|---|---|
| 1) Figure 1, there seems to be huge overlaps between Topt for different layers (25th-75th percentile box), how would this impact the results? Have the authors consider perhaps running the model woth those as upper and lowes limits on Topt to see the impact on performance? | The variability in $T_{opt}$ likely reflects the differences in environmental conditions and sensor depths between calibration sites, resulting in large overlaps in $T_{opt}$ IQR between product layers. Product comparison results presented in Figure 3 reaffirm the limited sensitivity of the EF to variations in T-parameter observed by others (Ford et al. 2014; Grillakis et al. 2021). This is especially apparent in case of E5L layer 7–28 cm, where the performance of all four RZSM product layers is very similar. Nonetheless, each of the RZSM product layers correlates best with its most-approximate E5L counterpart in all but one case. |
| 2) Typically EF is implemented in a normaliezed SM scale (SWI either by scaling from 0-1 using min/max or using soil characteristic properties). In the MS, it's not mentioend which expecific method was used (if at all). | The scaling of the input SM data between 0–1 and subsequent rescaling between the wilting point and field capacity values is usually done when dealing with datasets that express SM as the percentage of saturation rather than in volumetric units ($m^3/m^3$), such as in the EUMETSAT H SAF soil moisture data records. Given that the C3S SM data is already in volumetric units, there is no need for rescaling. |
| 3) Lines 45-48, could use more detailes on existing methods for rootzzone SM estiamtions and their challenges. | We will revise these lines to include more detail: "The existing link between SM dynamics in the surface layer and the root zone (Albergel et al., 2008; Wang et al., 2017; Ford et al., 2014; Sure and Dikshit, 2019) allows for estimating RZSM from surface SM (SSM) observations via a variety of hydrological models. These include relatively simple two-layer approaches approximating RZSM as a function of SSM (Manfreda et al., 2014), compound process-based models requiring sophisticated parameter calibration (Bouaziz et al., 2020), as well as complex and computationally expensive land surface models requiring many auxiliary inputs (Muñoz Sabater et al., 2021; Rodell et al., 2004). Satellite-based SSM observations can also be assimilated into land surface models to improve the model simulations of RZSM with global and temporally-complete coverage, as in the case of the SMAP L4 RZSM product (Reichle et al., 2017)." |

| | |
|---|---|
| 4)Line 69, EF typically is used to estimate SM at specific layer depth not a composites like 0-10 or 0-40. It's either 0 or 10 or 40 cm ($\pm$ few cms) | Both approaches are common practice. For example, in the aforementioned SMOS L4 product, the EF is used to represent a 5–40 cm soil layer (Al Bitar and Mahmoodi, 2020). Other studies have used 0–100 cm (Wagner et al. 1999, De Lange et al. 2008); 0–25, 50–100, and 0–100 cm (Ceballos et al. 2005); or 25–60 cm (Ford et al. 2014). Moreover, we believe that, given the large variations in land cover and soil texture within single satellite grid cells, it is impossible to assign a single specific depth to the EF-based RZSM estimates. Other practical considerations were also taken into account here, e.g., binning of the very limited number of in situ sensors operating at greater depths is necessary to obtain a reasonable sample against which to calibrate the T-parameter. Therefore, we think that a depth range accompanied by rigorous uncertainty estimates (which are the main goal of the study) provide a more realistic description of the product. |
| 5) Section 2.1, I would suggest to include a table showing the timeline of various satellite SM products being part of C3S with band information that would help understanding the dataset better. Also, I would liked to see some examples of mentioned structural breaks either as timeseries. | Individual sensor data are intercalibrated within the input C3S SM dataset and the breaks in the *SSM* signal occurring at sensor changes were demonstrated by Preimesberger et al. (2021) to be marginal. However, the structural breaks—in the new manuscript referred to as systemic breaks, see above—are distinct in the C3S product *uncertainties*. We will include a sensor timeline table in section **2.1 C3S surface soil moisture**. We will also clarify this in the MS (see our response to major comment #3). A time series example of a systemic break in C3S is shown in Figure 6. We will redesign Figure 6 to accommodate specific comment #9 and also to make the systemic breaks in C3S uncertainty time series clearly visible (revised Figure 6 is included at the end of this document). |

| | |
|---|---|
| 6) Figure 3, I dont think there is any need to compare all the layers at each depth. For instance, deeper layers should not be compared with 0-7 cm modeled SM. They are not the same thing to be comapared and does not add any value to the MS. Similarly, at 100-200 cm depth, surface SM correlations and their discussions (Lines 241-250) could be avoided. | Regarding Figure 3, the reason for comparing all of the RZSM product layers as well as the input C3S SSM layer against E5L is to demonstrate that our approach to T-optimization works and that the sensitivity of the EF method to variations in $T_{opt}$ is limited, as discussed in the MS earlier. The former is reflected in the best performance of each RZSM layer being achieved again at the depth of the E5L product we intended it to; the latter is apparent in the very similar performance of all RZSM product layers against E5L reference (regardless of significant differences in $T_{opt}$ between them). For these reasons we believe this figure provides more information in its current form and we would like to keep it. We will also annotate the median lines in every box for an easier comparison of the results (revised Figure 3 is included at the end of this document). |
| 7) Line 259, the structural uncertainty mentioned here is it same as the structural breaks (Line 87)? if not, perhaps use some other terminology to differentiate is further. | Model structural uncertainty refers to an error inherent in the EF method and is introduced and explained much earlier in L64-66 of the submitted MS (although in first instance referred to as model structural *error*, which we will change also to model structural *uncertainty* for consistency). Subsections 3.2.3 (in methods) and 4.3 (in results) discussing the model structural uncertainty were both given the same name to make it clear what is being referred to. Structural breaks are sudden changes in the input uncertainty data and are described when introducing C3S dataset (L87 of the submitted MS). We will use the term *systemic breaks* instead to better distinguish the two terms. Further clarification regarding the difference between the systemic breaks in the SSM signal and those in the uncertainty data will be provided as per our response to major comment #3. |
| 8) Line 268, please add exact location of the data. | Thank you for spotting the missing location. We will revise the caption to: *Figure 5a shows a time series of RZSM uncertainties from the baseline method at an arbitrary example location in Benin (9.875N, 1,625E).* |
| 9) Figure 6, very busy plot. Hard to read. | Thank you for the suggestion. We will redesign Figure 6 for more clarity by removing the not relevant in situ signal and placing uncertainties in separate panes with independent scales to make their temporal dynamics better visible (find the revised version at the end of this document). |

| 10) Figure 7, how is the sample size in the subset bigger than the whole dataset (Fig d vs h). | Thank you for pointing this out. The time series selected for this analysis were filtered for a minimum Pearson's r ($r \geq 0.5$) to mitigate the impact of the spatial mismatch between point-scale in situ measurements and the large footprint of the satellite observations (as described and referenced in L162-164 of the submitted MS). In this experiment, several time series that did not satisfy this minimum correlation criteria in the 2002-2020 period (7d) reached the $r \geq 0.5$ threshold in 2015-2020 (7h). This is due to the latter part of the time series being based on more modern satellite sensors providing more accurate SM retrievals. We will reprocess Figure 7 using the exact same sample, i.e., only sites where $r \geq 0.5$ in both periods. We do not expect this to impact the overall result in a significant way, though. |

[Figure]

Figure 3: Product intercomparison of the C3S SSM and RZSM products against E5L SM.

[Figure]

Spatial correlation maps of the C3S SSM (a) and RZSM products (b-e) with E5L layer 0–7 cm (a-b), 7–28 cm (c), 28–100 cm (d) and 100–289 cm (e).

[Figure]

Figure 6: Differences in uncertainty variations of the baseline (a-b) and our proposed uncertainty estimation approach (c-d). Illustrated on the example of RZSM layer 2 at an arbitrary location in Benin (9.875N, 1,625E).

---

## Author Comment (AC2)

**Responses to reviewer's comments**

We thank the editor and the reviewers for their time and effort to review our manuscript. Please find our replies to all comments below.

Reviewer comments are marked in red.
Responses to the comments are marked in blue.
Changes that will be made in the revised manuscript are marked in *italic*.

**Reviewer 2:**

The manuscript "Uncertainty estimation for a new exponential filter-based long-term root-zone soil moisture dataset from C3S surface observations" by A. Pasik et al. describes the development of a root-zone soil moisture (RZSM) dataset based on the C3S near-surface soil moisture using the exponential filter method. The newly derived product contains estimates of root-zone soil moisture for different depths, but also their uncertainty estimates considering several sources of uncertainty and their propagation in time. The data product and the code to generate this dataset are available online. I enjoyed reading this well written and clear paper. Please read below my comments which hopefully will help to further improve the manuscript.

We thank the reviewer for their positive feedback and constructive comments.

| Major Comments | Reply |
|---|---|
| 1. The root-zone soil moisture is defined here as the water present in the top meter of the soil column (p2, lines 38-39). However, I would argue that the root-zone soil moisture represents the water in the sub-surface which is accessible to the roots of the vegetation for transpiration. The depth thereof is highly variable and may depend on climatic (de Boer-Euser et al., 2016; Wang-Erlandsson et al., 2016) and topographic indicators (Fan et al., 2017). Now the derived product provides estimates of soil moisture at different depth intervals, but not really an integrated root-zone soil moisture estimation, which depends on the rooting depth. Would such an addition in the dataset be feasible? | We acknowledge the observation that root-zone soil moisture represents the water in the sub-surface which is accessible to the roots of vegetation for transpiration. We also appreciate the potential utility of the suggested integrated root-zone dataset, especially in land surface and hydrological modelling. We indeed plan to investigate the feasibility of producing such an integrated variable in the future, informed by a plant-rooting depth map like the one referenced here (Fan et al. 2017). This is however outside the scope of this study, which is primarily tasked with developing a comprehensive uncertainty estimation scheme for the exponential filter method. |

2. When comparing the newly derived product with ERA5L and local data, it would be interesting to see plots of timeseries at specific locations and a spatial map showing where both products have high or low correlations. This would give potential users of the dataset some guidance on where the product may and may not be used. Perhaps the authors can work out some correlations to relate skill with catchment characteristics including climate, land use, topography and soil types.

We will include spatial correlation maps between the C3S SSM and RZSM products with their respective ERA5-Land counterparts in an additional new Figure (please find it included at the end of this document). We believe this will give a spatially more complete understanding of the product's performance than time series could. We will discuss the observed spatial patterns: *"A global SM dataset spanning the 2002–2020 period was computed using the EF method and T-parameters optimized at the point scale with the approach described in section 4.1. Figures Xa–e) show correlation maps of each of the RZSM product layers as well as the input C3S SSM dataset with E5L. The spatial patterns observed in the C3S SSM data (Figure Xa) are strikingly similar to those in RZSM layer 1 (Figure Xb) with slight to moderate deterioration in performance over the high latitudes ($> 60°N$). This is not surprising given that both products differ only by a small degree of smoothing applied to RZSM layer 1 and are compared to the same E5L layer (0-7 cm). RZSM layers 2 and 3 (Figure Xc-d) are compared to E5L layers 7-28 and 28-100 cm, respectively, and largely preserve good performance in regions where the input C3S SSM product also performs well, i.e., in Europe (bar Scandinavia), the Caspian and Aral Sea basins, the Eastern United States, India, Southeast Asia, South America, Sub-Saharan Africa, and Australia. At the same time, deterioration of performance is observed in high latitudes and in arid environments such as the Sahara desert and the Arabian Peninsula where the reduced strength of coupling between the surface and root-zone dynamics may hinder the EF performance (Yang et al., 2022). The patterns of good and poor performance visible in RZSM layers 1-3, are not replicated in RZSM layer 4 (Figure Xe) where the agreement with the reference E5L 100-289 cm layer is spatially very heterogeneous and worse overall. The few regions where the good performance observed in shallower layers is preserved include India, Southeast Asia, and the Eastern United States"*.

The impact of soil type or land cover on the performance of the EF remains ambiguous (e,g., Stefan et al. 2021). External variables could potentially influence the uncertainty estimates, but such detailed analysis is outside of the scope of this study. We will reiterate the focus of this study throughout the MS, e.g., as per major comment #1 and also by adding the following concluding sentences to the MS' Introduction: *"The focus and novelty of this paper lie in quantifying, rather than reducing, the EF model's known limitations by providing a methodology for comprehensive uncertainty*

| Minor Comments | Reply |
| --- | --- |
| 3. line 64: I would specify here: "does not consider the model structural error of the EF method". Although this becomes clear later in the paper, it was not directly clear to me at this stage. | Thank you for the suggestion. We will change L64 to: *"This approach takes into account the uncertainties of both the SSM input data and the EF model parameter, but does not consider the model structural uncertainty (Beven, 2005) of the EF method."* |
| 4. line 85: "uncertainties [..] were then calculated from the law of propagation of uncertainties". Could you explain how this was done in more detail? | It is literally applying the law of the propagation of uncertainties to the merging equation (weighted average), i.e., $\sigma^2_{\varepsilon_m} = \sum_i w_i^2 \sigma^2_{\varepsilon_i}$, where $\sigma^2_{\varepsilon}$ is the error variance, $m$ refers to the merged SSM estimate, and $i$ refers to the input SSM products that are being merged. We will make this more clear by extending the sentence in question to: *"Uncertainties of the merged SSM estimates were then calculated from the law for the propagation of uncertainties (i.e., predicting the uncertainty reduction due to the weighted averaging, assuming that merging weights are correct; see Gruber et al. (2017))"* |

| | |
|---|---|
| 5. section 3.2.3: could you explain the presented formulas in more detail? What are the units and what are all parameters? E.g. what does G represent? What does delta represent? | Thank you for this suggestion, we believe the reviewer is referring to section 3.2.1. We will add further details to this section to make the formulas and parameters more understandable. *"In De Santis and Biondi (2018), the standard law for the propagation of uncertainties is applied to the EF method, assuming the errors in the SSM inputs and T-parameter to be normally distributed and uncorrelated. We use this approach as a baseline for our analyses. The recursive formulation of this baseline method is as follows:* **(equations 4–7)** $\sigma(RZSM)$ *and* $\sigma(T)$ *denote the uncertainty of the RZSM estimates (in* $m^3/m^3$*) and the EF model parameter* $T$ *(a unit of time, in days), respectively. The equation is initialized as* $\Delta_0 = \sigma(SSM_0)$*,* $\partial RZSM_0 / \partial T = 0$ *and* $G_0 = 0$*. Uncertainties of the SSM input data are considered by the* $\Delta$ *term (in* $m^3/m^3$*), which also takes into account the effect of possible prolonged input data gaps dependent on the* $T$*-value. The Jacobian term* $\partial RZSM/\partial T$ *assumes high values proportional to the latest SSM input variability on a time scale related to the* $T$*-parameter (expressed as* $m^3/m^3$ *over time). This is reflected in significant changes in the RZSM value associated with wetting or drying of the soil. Finally, the term* $G$ *(dimensionless) weighs the contribution of change recorded between the latest and penultimate RZSM estimates"*. |
| 6. line 246: could you quantify with numbers in the text how substantial the difference is between the correlation between E5L and RZSM versus E5L and SSM? | The correlation values for E5L/SSM/RZSM are discussed in L242-249 of the submitted MS. We will annotate the median lines in each box in Figure 3 (revised figure included at the end of this document) for a better overview of differences in correlation scores. |
| 7. Figure 5: in the legend I read "GCOS required uncertainty" but it is not entirely clear to me what this threshold refers to exactly, could you please elaborate? | GCOS outlines target accuracy requirements for ECV data products that are determined by the scientific community. These requirements can be found in the so-called "GCOS Implementation Needs" (GCOS, 2022). We will add a clarifying statement to the manuscript: *"The dashed grey line indicates the uncertainty level defined by GCOS (2022) as an accuracy goal for RZSM products."* |

| | |
|---|---|
| 8. line 280: could you elaborate more on why we expect uncertainties to be amplified during transitions between wet and dry conditions? Which processes play a role which are not well represented in the EF method? Now you briefly refer to Fig6, but it does not provide a clear explanation on why this is expected. | Essentially, the EF method operates by smoothing the variations in the SSM signal, therefore the sudden changes are attenuated and delayed while in reality they can be more significantly transmitted to the deeper layers. This simplistic nature of the model results in its limited ability to capture the wetting/drying accurately and, for that reason, in larger uncertainties. We will further clarify this by adding a sentence following the one referred to here. Together, this will read: *"Compared to the baseline (Figure 5a), this yields an increased overall magnitude of the uncertainties, a more realistic increase in (temporal average) uncertainties with depth, and an amplified temporal variability in all layers during transitions between dry and wet conditions (see Figure 6). The latter effect is caused by the simplistic nature of the model, which essentially operates as a smoother and therefore attenuates sudden variations in the SSM signal which in reality may be transmitted into the deeper layers in a more significant manner. The reduced accuracy of the EF method during soil wetting and drying phases was also observed by others (Ford et al. 2014)."* |
| 9. line 292 "and highlighting 20% of data with the highest uncertainty". At this stage, this reads a bit confusing as the previous paragraph describes masking out data with the highest uncertainty and here (if I understood correctly) you are instead plotting data with high uncertainty. Perhaps good to clarify what you mean with "highlighting 20% of data with the highest uncertainty". | We will edit the referenced sentence (L303-304) to read: *Figure 6a) and d) indicate (in magenta shading) 20% of RZSM layer 2 data with the highest uncertainties masked out in the experiment described above based on uncertainties estimated with the baseline (b), and our method (d), respectively.* |
| 10. line 299: the described difference in uncertainty from 0.008 m3/m3 to 0.004 m3/m3 is very hard to see in the Figure using the applied scale. | Figure 6 will be redesigned for more clarity by removing the not relevant in situ signal and placing uncertainties on independent scales where their temporal dynamics are better visible (please see its revised version at the end of the document). |

| | |
|---|---|
| 11. line 302: why are the uncertainties related to structural breaks not clearly seen in the MAD Topt approach. Can you reflect (here or later in the discussion) a little bit more on this result. It seems to me that the change in uncertainty related to a change in sensor is an important change that you would also want to see in the improved methodology for uncertainty estimation. | Thank you for this comment. Indeed, the uncertainties of the input data are an important element here and their sudden shifts should be reflected in the propagated values. We observe that in this particular aspect, the difference between the baseline and our methods is driven solely by the value assumed by the $\sigma(T)$. A higher value of $\sigma(T)$ places more weight on the impact of significant changes in RZSM values (represented by the Jacobian term $\partial RZSM_0 / \partial T$), while lower $\sigma(T)$ favors the impact of the input uncertainty values ($\Delta$). The former better reflects the sudden changes in input uncertainty due to sensor changes, while the latter is more suited to resolving day-to-day uncertainty variations. This seems to be somewhat of a trade-off with this approach as it cannot do both at the same time. We will extend the discussion of this result in Section 5 **Summary and Conclusions** by adding the following sentences to the existing comparison of the used $\sigma(T)$ values (L335-340 of the submitted MS): *"A higher value assumed by $\sigma(T)$ (in this case $MAD(Topt)$) places more weight on short-term significant variations in RZSM values (accounted for by the Jacobian term $\partial RZSM/\partial T$) and overshadows the contribution of the input uncertainties ($\Delta$) to the overall uncertainty budget. This approach results in higher uncertainty outputs paralleling significant changes in RZSM signal (e.g., soil wetting/drying events) and is generally better suited to describe day-to-day uncertainty variations. Meanwhile, lower value of $\sigma(T)$ (here $Topt/10$) favors the impact of the input uncertainties and appears to be more skillful in detecting sudden shifts in the magnitude of the input uncertainties due to C3S SSM sensor changes. While both the significant variations in RZSM values and the magnitude shifts in the input uncertainties are crucial elements of the overall uncertainty budget, there appears to be a trade-off in favoring the impact of one or the other based on the value assumed by $\sigma(T)$"*. |

| | |
|---|---|
| 12. line 314-320: here, it is not clear to me why using Topt/10 as T parameter uncertainty yields more realistic estimates of temporal uncertainty variations than using MAD Topt in the case of using the time series which includes a structural break (and the opposite in case a shorter time series is used). Which aspects in figure 7 suggest these findings? | In fact, both approaches yield realistic temporal variations as they mostly assign highest uncertainty values to the same RZSM estimates (this is evident in Figure 6). In the context of Figure 7, we describe the uncertainties which yielded better results in improving R with the in situ reference as *"more realistic estimates of temporal uncertainty variations"*. Indeed in reference to Figure 7 this sentence could be more clear. We will rephrase it: *"In case of the full product period (Figure 7a–d), using $\sigma(T) = Topt/10$ as T parameter uncertainty seems to yield more consistent improvements in correlation with the in situ reference after removing a percentage of the most uncertain data, than using $\sigma(T) = MAD(Topt)$."* The so called structural breaks are sudden shifts in the magnitude of the input uncertainties and the value assumed by $\sigma(T)$ either increases or decreases the weight of their contribution to the total uncertainty budget. Please see our response to the previous comment (#11). |
| 13. line 320-324: Again, could you elaborate why the uncertainty estimations of temporal uncertainty variations are no longer accurate for deeper layers? | Thank you for the comment, we will elaborate on the challenges of capturing the temporal uncertainty variations in deeper layers by including the following sentences at the end of section 3.5 **Assessment of uncertainty estimates**: *"At greater depths, the contribution of the model structural uncertainty on the total uncertainty budget has been shown to increase. In the circumstances where the EF model appears to be inadequate, for example due to poor coupling between the root zone in consideration and the surface layer, it can be assumed that the model structural uncertainty is so predominant as to make the temporal patterns of the other uncertainty components marginal in practice. However, in circumstances where the magnitude of the real uncertainty is such as to make the EF-based RZSM so unreliable, the lack of ability to reproduce the temporal variations of the estimated uncertainty becomes less relevant."* |
| 14. line 351: here, you forgot to add the units of the mentioned uncertainties. | Thank you for pointing this out, we will add the missing units. |

| | |
|---|---|
| 15. In addition, I also downloaded the netcdf files and checked the github page. The nc files contain all the necessary meta data. However, the github page does not include extensive documentation on how to use the different methods within the package. Would it be feasible to elaborate on this further? | We appreciate the thoroughness. While the package includes basic examples of the application of the package, we agree that a more extensive documentation would be beneficial and intend to expand it in the future. We made the code package public also in hopes that it will attract contributions from the user community. |

References:

Beven, K.: On the concept of model structural error, Water Science and Technology, 52, 167–175, https://doi.org/10.2166/wst.2005.0165, 2005.

de Boer-Euser, T., McMillan, H. K., Hrachowitz, M., Winsemius, H. C., & Savenije, H. H. G. (2016). Influence of soil and climate on root zone storage capacity. Water Resources Research, 52, 2009–2024. https://doi.org/10.1002/2015WR018115

De Santis, D. and Biondi, D.: Error Propagation from Remotely Sensed Surface Soil Moisture Into Soil Water Index Using an Exponential Filter, in: HIC 2018. 13th International Conference on Hydroinformatics, edited by Loggia, G. L., Freni, G., Puleo, V., and Marchis,M. D., vol. 3 of EPiC Series in Engineering, pp. 520–525, EasyChair, https://doi.org/10.29007/kvhb, 2018.

Fan, Y., Miguez-Macho, G., Jobbagy, E., Jackson, R. and Otero-Casal, C. (2017) Hydrologic regulation of plant rooting depth, PNAS, 114 (40), 10572-15077, doi.org/10.1073/pnas.1712381114

Ford, T. W., Harris, E., and Quiring, S. M.: Estimating root zone soil moisture using near-surface observations from SMOS, Hydrology and Earth System Sciences, 18, 139–154, https://doi.org/10.5194/hess-18-139-2014, 2014.

GCOS: The 2022 GCOS ECVs Requirements, World Meteorological Organisation, 245, https://public.wmo.int/en/resources/library/global-observing-system-climate-implementation-needs, 2022.

Gruber, A., Dorigo, W., Crow, W., and Wagner, W.: Triple Collocation-Based Merging of Satellite Soil Moisture Retrievals, IEEE Transactions on Geoscience and Remote Sensing, PP, 1–13, https://doi.org/10.1109/TGRS.2017.2734070, 2017.

Wang-Erlandsson, L., Bastiaanssen, W. G. M., Gao, H., Jägermeyr, J., Senay, G. B., Van Dijk, A. I. J. M., et al. (2016). Global root zone storage capacity from satellite-based evaporation. Hydrology and Earth System Sciences, 20(4), 1459–1481. https://doi.org/10.5194/hess-20-1459-2016

Yang, Y., Bao, Z., Wu, H., Wang, G., Liu, C., Wang, J., Zhang, J. An Exponential Filter Model-Based Root-Zone Soil Moisture Estimation Methodology from Multiple Datasets. Remote Sens. 2022, 14, 1785. https://doi.org/10.3390/rs14081785

[Figure]

Figure 3: Product intercomparison of the C3S SSM and RZSM products against E5L SM.

[Figure]

Spatial correlation maps of the C3S SSM (a) and RZSM products (b-e) with E5L layer 0–7 cm (a-b), 7–28 cm (c), 28–100 cm (d) and 100–289 cm (e).

[Figure]

Figure 6: Differences in uncertainty variations of the baseline (a-b) and our proposed uncertainty estimation approach (c-d). Illustrated on the example of RZSM layer 2 at an arbitrary location in Benin (9.875N, 1,625E).